# Effects of Mono-Vacancies of Oxygen and Manganese on the Properties of the MnO_2_/Graphene Heterostructure

**DOI:** 10.3390/ma15082731

**Published:** 2022-04-08

**Authors:** Juan David Morinson-Negrete, César Ortega-López, Miguel J. Espitia-Rico

**Affiliations:** 1Grupo Avanzado de Materiales y Sistemas Complejos GAMASCO, Universidad de Córdoba, Montería CP 230001, Colombia; juanmorinson@correo.unicordoba.edu.co (J.D.M.-N.); cortegal@correo.unicordoba.edu.co (C.O.-L.); 2Doctorado en Ciencias Física, Universidad de Córdoba, Montería CP 203001, Colombia; 3Grupo de Investigación AMDAC, Institución Educativa José María Córdoba, Montería CP 230001, Colombia; 4Grupo GEFEM, Universidad Distrital Francisco José de Caldas, Bogotá CP 110111, Colombia

**Keywords:** T–MnO_2_/graphene heterostructure, Bader charge, magnetic metal, DFT

## Abstract

The effects of the monovacancies of oxygen (V_O_) and manganese (V_Mn_) on the structural and electronic properties of the 1T–MnO_2_/graphene heterostructure are investigated, within the framework of density functional theory (DFT). We found that the values of the formation energy for the heterostructure without and with vacancies of V_O_ and V_Mn_ were −20.99 $\frac{\mathrm{meV}}{{Å}^{2}}\;$, −32.11$\frac{\mathrm{meV}}{{Å}^{2}}$, and −20.81 $\frac{\mathrm{meV}}{{Å}^{2}}$, respectively. The negative values of the formation energy indicate that the three heterostructures are energetically stable and that they could be grown in the experiment (exothermic processes). Additionally, it was found that the presence of monovacancies of V_O_ and V_Mn_ in the heterostructure induce: (a) a slight decrease in the interlayer separation distance in the 1T–MnO_2_/graphene heterostructure of ~0.13% and ~1.41%, respectively, and (b) a contraction of the (Mn−O) bond length of the neighboring atoms of the V_O_ and V_Mn_ monovacancies of ~2.34% and ~6.83%, respectively. Calculations of the Bader charge for the heterostructure without and with V_O_ and V_Mn_ monovacancies show that these monovacancies induce significant changes in the charge of the first-neighbor atoms of the V_O_ and V_Mn_ vacancies, generating chemically active sites (locales) that could favor the adsorption of external atoms and molecules. From the analysis of the density of state and the structure of the bands, we found that the graphene conserves the Dirac cone in the heterostructure with or without vacancies, while the 1T–MnO_2_ monolayer in the heterostructures without and with V_O_ monovacancies exhibits half-metallic and magnetic behavior. These properties mainly come from the hybridization of the 3d–Mn and 2p–O states. In both cases, the heterostructure possesses a magnetic moment of 3.00 *μ_β_*/Mn. From this behavior, it can be inferred the heterostructures with and without V_O_ monovacancies could be used in spintronics.

## 1. Introduction

The successful exfoliation of graphene in 2004 [1] unleashed the search for new bidimensional materials, so monolayers such as those of boron nitride, oxides, dioxides, carbides, and carbonitrides of transition metals are a reality today [2,3,4,5,6,7,8]. Within these new monolayers, some theoretical studies have been centered on a monolayer of MnO_2_, finding that it crystallizes in the prismatic 1H (D_3h_) and trigonal 1T (D_3d_) phases [9,10]. A MnO_2_ monolayer was successfully synthesized for the first time in 2003 by Omomo et al. [11]. Subsequently, this monolayer has been grown using different experimental methods [12,13,14,15,16]. On the other hand, by means of theoretical studies using DFT, Kan et al. demonstrated that the MnO_2_ monolayer exhibits indirect and ferromagnetic semiconductor behavior with a bandgap of 3.5 eV and a total magnetic moment of 3.0 *µ_β_*/cell [17]. In addition, various theoretical and experimental studies have explored the effects of O and Mn vacancies on the properties of a MnO_2_ monolayer [18,19,20,21,22]. For example, Wang et al. [23] found that the inclusion of O and Mn vacancies causes significant variations in the semiconductor nature of the monolayer, acquiring a robust half-metallic behavior, while Sakai et al. [24], through experimental studies, demonstrated that a layer of MnO_2_ with Mn vacancies generates photocurrents under visible light irradiation. Therefore, the inclusion of this type of precise defects generates new properties in the monolayer and opens the door for new applications. Regarding graphene, since it was obtained it has been intensively studied both theoretically and experimentally. Over the last few years, there has been great interest in the use of graphene in spintronics, but pure graphene is nonmagnetic. Some methods such as doped graphene 3d transition metals and the generation of defects such as vacancies in graphene have been used to overcome this difficulty. For example, via DFT, Ostovari et al. [25] studied graphene in the presence of Fe dopant and vacancy; they found both the generation of a vacancy and the substitution with an Fe atom induce magnetic effects in graphene, with the FM phase being the most probable.

On the other hand, with the aim of increasing the potential applications of monolayers, many researchers have centered their studies on the joining of bidimensional materials, forming heterostructures that allow combining the properties of two or more 2D materials [26,27,28,29,30,31,32,33,34,35,36]. With respect to the MnO_2_/graphene heterostructure, it has been studied theoretically [37,38,39,40]. These investigations showed that the heterostructure can be used as a good anode material for lithium-ion batteries and for electrochemical energy storage in batteries. The heterostructure has been grown using different experimental methods [39,40,41,42,43]. In addition, some theoretical studies, such as that of Gan et al. [44], have found that due to the interfacial charge transfer from graphene, a half-metallic behavior is induced in the heterostructure. This suggests that the MnO_2_/graphene heterostructure could be important in the field of spintronics. On the other hand, some experimental studies have demonstrated that the MnO_2_/graphene heterostructure exhibits an excellent performance in various technological and environmental applications, such as energy storage in batteries [40], supercapacitors [41,42], and the removal of environmental contaminants such as formaldehydes [45] and tetracycline in pharmaceutical residue water [46]. Nevertheless, although great effort has been made to establish the potential for applications of this heterostructure, presently studies related to the effects of O and Mn vacancies on the structural, electronic and magnetic properties of the MnO_2_/graphene heterostructure and its energy stability have not been reported. For this reason, it would be interesting to conduct a theoretical study of the effects of the inclusion of O and Mn monovacancies on the structural, electronic, and magnetic properties of the 1T–MnO_2_/graphene heterostructure. Because the vacancies of oxygen and manganese in the heterostructure could modify the electronic properties and generate active sites that improve the adsorption of lithium atoms and environmental contaminants compared to the performance of the heterostructure without vacancies.

## 2. Computational Method

The calculations were carried out within the framework of density functional theory (DFT) [47,48]. In this theory, it is established that the expected value of any observable of the base state is a single functional of the electronic density of the base state; in particular, the total energy of the system of many electrons and ions is a density functional *E* = *E*[*ρ*(*r*)].

Here, the electronic density *ρ*(*r*) is defined as the number of electrons per unit of volume. The MnO_2_/graphene heterostructure is a system made up of many electrons and many ions, whose movements are coupled by the coulomb interaction that exists between them and by non-classical interactions (correlation exchange) between the electrons. In order to construct a theoretical model that describes this system, it is necessary to solve a quantum mechanical problem that consists of the interactions of a large number of particles (in the order of the Avogrado number, 6.023 × 10^23^).

With the introduction of the Born–Oppenheimer and the adiabatic approximations, the separation of the electronic degrees of freedom from the ionic degrees of freedom and the absence of transitions between electronic states, respectively, is guaranteed. This leads us to solving the time-independent Shrödinger equation or for stationary states. The Kohn–Sham (KS) equations allow us to map a system of many interacting electrons and ions by means of a system of non-interacting electrons as long as their electronic densities are the same were solved using the Quantum-ESPRESSO computational code [49,50]. The exchange–correlation potential was modelled on the generalized-gradient approximation (GGA) in the Perdew–Burke–Ernzerhof (PBE) parametrization [51], while the external potential was described by means of ultrasoft pseudopotentials. The Kohn–Sham electronic orbitals were expanded on a plane wave basis with a cutoff kinetic energy for the wave and charge density functions of 50 and 500 Ry, respectively. The heterostructures with and without vacancies were modelled using the periodic slab method; in these calculations, we did not saturate the carbon atoms with hydrogen atoms, because we used a large empty region of 20 Å in order to avoid interactions between the heterostructure and its image. In order to include the effects of the weak interaction between the 1T–MnO_2_ monolayer and the graphene, the D2 approximation was used [52]. All of the structural relaxation calculations were carried out taking into account the convergence criteria for the total energy, the force, and the pressure (variable cell calculations) of 1 meV/atom, 1 meV/Å, and 0.2 kbar, respectively.

The 1T–MnO_2_/graphene heterostructure was modelled on the vertical growth of a monolayer of 1T–MnO_2_ on graphene. As can be seen, the mismatch between these lattice parameters is relatively large; that is to say, it is necessary to look for configurations in which the mismatch is less than 3%. There are infinite configurations that meet this requirement, but in this paper the 1T–MnO_2_ ($2\sqrt{3}\times 2\sqrt{3}$)/graphene (4×4) configuration will be used with the Mn centered on the graphene hexagon and a mismatch of 1.33%. The choice of this configuration is based on the fact that this heterostructure is slightly more stable than the configurations with O centered on the graphene hexagon [44]. Furthermore, this configuration is useful for modelling concentrations of vacancies per molecule of MnO_2_ of 8.3%.

The integrations in the first Brillouin zone in the calculations of the electronic structures were carried out in a 14×14×1 Monkhorst-Pack grid [53], while the study of the partial occupation of the electronic states near the Fermi level (*E_F_*) was conducted with the Methfessel–Paxton technique [54]. Finally, the calculations of the Bader pseudo-charge transfer [55] were carried out using the computational code for Bader charge created by the Henkelman group [56].

## 3. Results and Discussions

In this section, we present the results and discussions of the effects produced by the structural, electronic, and magnetic properties of the T–MnO_2_/graphene heterostructure caused by the neutral monovacancies of O (V_O_) and Mn (V_Mn_).

### 3.1. Strucural Parameters

Initially, the lattice constant of the isolated monolayers was optimized, and we obtained 2.882 Å for the 1T–MnO_2_ and 2.463 Å for the graphene. These results are in excellent agreement with the theoretical values reported by Rasmussen et al. [4], 2.89 Å for 1T–MnO_2_, and Hernandez et al. [10] and Espitia et al. [57], 2.46 Å for graphene; the discrepancies are less than 1%. However, the calculated values in this investigation are slightly higher than those experimentally reported by Fekuda et al. [12], 2.85 Å 1T–MnO_2_, and Novoselov et al. [1], 2.45 Å for graphene. These slightly higher discrepancies with respect to the experimental values occur because, as is known and accepted, the GGA approximation overestimates the values of the lattice constants.

With the aim of including the monovacancies, the configurtion of the heterostructure considered in this investigation has the general form T−${\mathrm{Mn}}^{12-x}O_{2}^{24-y}$($2\sqrt{3}\times 2\sqrt{3}$)/graphene (4×4), where x and y represent the number of neutral V_Mn_ and V_O_ monovacancies and the exponents (12−x) and (24−y) correspond to the total number of Mn and O atoms, respectively. As was previously mentioned, the vertical stacking pattern considered for the heterostructure is an atom of Mn centered on the graphene hexagon. The heterostructure is made up of sixty-eight (68) atoms (32 C, 24 O, and 12 Mn) distributed on four horizontal planes: an upper plane of O atoms, a plane of Mn atoms, a lower plane of O atoms, and finally the graphene. Figure 1 shows top and side views of the 1T–MnO_2_/graphene heterostructures without vacancies and with a vacancy of V_O_ and V_Mn_. The orange, blue, and yellow spheres represent the O, Mn, and C atoms, respectively. Figure 1a corresponds to the heterostructure without vacancies (x=0 and y=0), and Figure 1b,c illustrate the heterostructures with V_O_ (x=0 and y=1) and V_Mn_ (x=1 and y=0). The red and blue circles in Figure 1b,c represent the V_O_ and V_Mn_ vacancies, respectively.

With the aim of studying the energy stability of the heterostructure without vacancies and with V_O_ and V_Mn_ vacancies, the formation energy (*E_f_*) was calculated.

The formation energy was calculated using the equation [58]:(1)$$E_{f}=\frac{E_{h}-E_{m}^{\left(1\right)}-E_{m}^{\left(2\right)}}{A}$$
where $E_{h}$ corresponds to the total energy of the heterostructures, $E_{m}^{\left(1\right)}$ and $E_{m}^{\left(2\right)}$ are the total energy of the isolated monolayers that make up the heterostructures, and A is the area of the horizontal plane of the heterostructures.

The values calculated for the formation energy were $E_{f}^{no\;vacancy}=-20.81\;\frac{\mathrm{meV}}{{Å}^{2}},\;E_{f}^{V_{O}}=-20.99\;\frac{\mathrm{meV}}{{Å}^{2}}\;\mathrm{and}\;E_{f}^{V_{Mn}}=-32.11\;\frac{\mathrm{meV}}{{Å}^{2}}$. Given that each formation energy is negative, the heterostructure without and with V_O_ and V_Mn_ vacancies is energetically stable. These results indicate that it is possible to grow the heterostructures.

In order to analyze the effects of the V_O_ and V_Mn_ vacancies on the structural properties of the heterostructure, the interfacial distance D is introduced, which defines the separation between the T–MnO_2_ monolayer and the graphene. D is calculated for the heterostructure with vacancies and is compared with the interfacial distance without vacancies. Additionally, in order to determine the variation in the bond lengths between the atoms that are first neighbors of the vacancy, the atoms are labeled from 1 to 15. In this way, the bond lengths between an atom of Mn and of O in general are denoted as *l_i-j_* (the distance between atoms *i* and *j*). These *l_i-j_* lengths were calculated for the heterostructure with V_O_ and V_Mn_ vacancies and were compared with the *l_i-j_* lengths of the heterostructure without vacancies.

In Table 1 and Table 2, the interfacial distance D and the *l_i-j_* bond lengths are listed for the T–MnO_2_/graphene heterostructure without vacancies and with V_O_ and V_Mn_ vacancies, respectively.

As can be seen in Table 1 and Table 2, the interfacial distance is 2.981 Å for the heterostructure without vacancies, while the interfacial separation values are 2.977 Å and 2.939 Å for the V_O_ and V_Mn_ vacancies, respectively. It should be noted that due to the V_O_ and V_Mn_ vacancies, a slight approachment between the T–MnO_2_ monolayer and the graphene is generated, were being ~0.13% for the V_O_ vacancy, while the relative decrease in D was ~1.41% for the V_Mn_ vacancy. Considering the l_i-j_ bond lengths between the i atom and the j atom next to the V_O_ and V_Mn_ vacancies, we found that there is a strong contraction between them, with this contraction being ~2.34% for the V_O_ vacancy, while the contraction for the V_Mn_ vacancy is greater, ~6.83%. The decrease in the l_i-j_ bond length is caused by the charge redistribution in the neighboring atoms of the V_O_ and V_Mn_ vacancies. We found that the V_Mn_ vacancy produces more significant structural changes in the heterostructure than the V_O_ vacancy, because of the fact that the Mn vacancy generates greater variations in the charge redistribution to the atoms that are first neighbors to this vacancy, which occurs because, as can be seen in Figure 1, the Mn atom is directly bonded to six (6) O atoms, while the O atom only has three (3) Mn atoms as first neighbors. In the following section, we will address this redistribution or electronic charge transfer in more detail.

### 3.2. Electronic Properties

The electronic and magnetic properties of the interface with neutral V_O_ and V_Mn_ monovacancies were studied by means of the calculation of the Bader charge, the band structure, and the density of states DOS.

#### 3.2.1. Bader Charge Transfer

The Bader charge transfer in the first and second neighbors of the V_O_ and V_Mn_ monovacancies were calculated and compared to the heterostructure without vacancies, and the variations of the Bader charge are calculated for the atoms labeled from 1 to 15, shown in Figure 1.

The variations of the Bader charge were calculated by means of the following equation:(2)$$\Delta Q_{i}=Q_{i}^{V_{X}}-Q_{i}^{no\;vacancy}$$
where $\Delta Q_{i}$ is the change in the Bader charge in the atom labeled with number *i*, X = O, Mn, $Q_{i}^{V_{X}}$ is the Bader charge in atom *i* near the O or Mn vacancy, and $Q_{i}^{no\;vancancy}$ is the charge of the same *i* atom without a vacancy. In Equation (1), changes in the charge with values of positive signs mean an excess of electrons, while negative values mean deficiencies of electrons.

In Table 3 and Table 4, the calculations of the changes in the Bader charge are shown in the first and second neighbors of the V_O_ and V_Mn_ vacancies, respectively.

The MnO_2_ monolayer in the 1T phase has an octahedral MnO_6_ structure. The Mn atom is bonded to six O atoms, and given that the O atom is more electronegative than the Mn atom, this Mn atom transfers charge to the six O atoms. When the V_O_ vacancy is generated (see Figure 1b), the Mn atom transfers charge to only five O atoms, which explains why the Mn atoms (labeled 2, 4, and 6) nearer the V_O_ vacancy have positive Bader charge variations, these values being $\Delta q_{2}=+0.214$, $\Delta q_{4}=+0.213,\;$ and $\Delta q_{6}=+0.205$. For the second neighbors, the variations were $\Delta q_{1}=+0.053$, $\Delta q_{3}=+0.047,$ and $\Delta q_{5}=+0.048$. These values are positive, but much lower than the values for the 2, 4, and 6 atoms, because atoms 1, 3, and 5 are not bonded to the O atom that was removed from the heterostructure in order to generate the vacancy. In accordance with the above, in the region around the V_O_ vacancy, important changes are generated in the interatomic charge transfer. This result is important, because these changes can generate chemically active regions that facilitate the adsorption of atoms and/or molecules.

On the other hand, for the V_Mn_ vacancy, the charges of the six O atoms (labeled 3, 7, 9, 11, 13, and 15 in Figure 1c) that are bonded to the Mn atom are significantly affected, due to the fact that they will not receive the additional charge of the Mn atom that was removed. As is shown in Table 4, there is a decrease in the Bader charges of the six first neighbors of the Mn vacancy, with the variations of the Bader charge being: $\Delta q_{3}=-0.157$, $\Delta q_{7}=-0.188$, $\Delta q_{9}=-0.156$, $\Delta q_{11}=-0.189$, $\Delta q_{13}=-0.150,$ and $\Delta q_{15}=-0.189$. The results for the Bader charge values explain the strong contraction that occurs between the six oxygen atoms and the Mn atoms labeled 2, 4, 8, 10, 12, and 14 in Figure 1c. Due to the fact that the six oxygen atoms are also bonded to these Mn atoms and since they do not receive charge from the Mn that was removed, these O atoms are strongly attracted toward the Mn atoms 2, 4, 8, 10, 12, and 14 in Figure 1c. This occurs because the Mn atoms (1.55) possess lower electronegativity than the O atoms (3.44).

Similarly, in the case of the V_O_ vacancy, these changes can generate chemically active regions that facilitate the adsorption of atoms and/or molecules.

#### 3.2.2. Electronic Properties

In order to study the electronic properties of the heterostructure, we calculated the band structure and the density of states (DOS) of the heterostructure without vacancies (1T–MnO_2_/graphene) and with V_O_ and V_Mn_ monovacancies. The results are shown in Figure 2. In the three cases, the Fermi level was chosen as the zero energy.

Figure 2(Ia,Ib) show, respectively, the band structure and the DOS of the heterostructure without vacancies. We found that the 1T–MnO_2_/graphene heterostructure exhibits metallic behavior. The behavior is caused by the graphene, due to the fact that the 2p–C states cross the Fermi level, just as the DOS in Figure 2(Ib) confirms, where it can be seen that the contribution of the 2p–C states (green line) crosses the Fermi level. In addition, it can be seen that the electronic properties of the graphene are well preserved. This is due to the fact that the graphene retains the Dirac cone, which is located on point K but is shifted ~0.7 eV in the conduction band, which explains the metallic character of the graphene in the heterostructure, while the 1T–MnO_2_ monolayer loses its previously reported semiconductor behavior [17] and acquires a half-metallic behavior, with the spin-up states being semiconductors while the spin-down states are metallic. A similar result was obtained by Gan et al. [44] in their first-principles study without vacancies of the two-dimensional MnO_2_/graphene interface. The DOS in Figure 2(Ib) confirms the half-metallic character of the monolayer in the heterostructure, where it can be seen that the greatest contribution to this behavior comes from the states with down spin 3d–Mn and the 2p–O states in lesser proportion, which cross the Fermi level. Finally, we found that the 1T–MnO_2_/graphene heterostructure without vacancies has magnetic properties, with a 3.0 *μ_β_*/Mn-atom magnetic moment, which comes from the hybridization of the 3d–Mn and 2p–O states.

Figure 2(IIa,IIb) show the band structure and the DOS of the heterostructure with an O vacancy, respectively. We found that the 1T–MnO_2_/graphene heterostructure with a (V_O_) vacancy preserves its metallic behavior, which is mainly due to the 2p–C orbitals of the graphene. This behavior is corroborated by the DOS in Figure 2(IIb), where it can be seen that the contribution of the 2p–C states (green line) crosses the Fermi level. In addition, we found that the graphene preserves the Dirac cone, and its location does not undergo significant changes with respect to the heterostructure without vacancies (Figure 2(Ia)), which is a consequence of the weak Van der Waals-type interaction between the 1T–MnO_2_ monolayer and the graphene. With respect to the 1T–MnO_2_ monolayer, we found that it preserves its half-metallic behavior, which, as the DOS in Figure 2(Ib) shows, is produced by the 3d–Mn states with a greater contribution and the 2p–O states in a smaller proportion. Additionally, we note that due to the V_O_ vacancy in the valence band near the Fermi level, there appear new full states mainly associated with the 3d–Mn orbitals with down spin. The figure of the band structure confirms this, where these bands appear closer together and almost parallel (see Figure 2(IIa)). These new full states appear because when the O vacancy is generated, the Mn atoms will transfer charge to only five O atoms and not to six, as occurs in the heterostructure without vacancies. As for the magnetic properties, we found that the 1T–MnO_2_/graphene heterostructure with V_O_ preserves the magnetic properties acquired in the heterostructure without vacancies. The heterostructure with V_O_ possesses a magnetic moment of 3.0 *μ_β_*/Mn-atom, which comes from the hybridization of the 3d–Mn and 2p–O states.

Finally, Figure 2(IIIa,IIIb) show the band structure and the DOS of the heterostructure with a Mn vacancy, respectively. We found that the 1T–MnO_2_/graphene heterostructure with a V_Mn_ vacancy retains its metallic behavior, which is caused by the 2p–C states of the graphene, which cross the Fermi level. Additionally, the Dirac cones remain intact, as is shown in Figure 2(IIIa). As for the 1T–MnO_2_ monolayer, it can be seen that it preserves its half-metallic behavior, but in contrast to what occurs in the 1T–MnO_2_/graphene heterostructure with a V_O_ vacancy, in which the main contribution to the half-metallic behavior comes from the 3d–Mn states with spin down; in the case of the 1T–MnO_2_/graphene heterostructure with a V_Mn_ vacancy, this contribution decreases and the half-metallic behavior of the monolayer comes from the 2p–O and 3d–Mn states with spin up with same contributions (see Figure 2(IIIb)). This behavior can be understood in the following way: when the Mn vacancy is generated, the six O atoms (labeled 3, 7, 9, 11, and 15 in Figure 1c do not receive the charge of the Mn atom, and furthermore, the contribution of the Mn atoms to the half-metallic behavior of the monolayer will decrease, since a Mn atom was removed. Finally, we found that the 1T–MnO_2_/graphene heterostructure with a V_Mn_ vacancy retains its metallic behavior, which is caused by the 2p–C states of the graphene that cross the Fermi level, and the 1T–MnO_2_/graphene heterostructure with a V_Mn_ vacancy possesses a magnetic moment of 2.72 *μ_β_*/Mn-atom. We observe that the magnetic moment decreases with respect to the heterostructure without vacancies and with a V_O_ vacancy, which happens upon removing a Mn atom, having in this case a lower contribution to the magnetism of the 3d–Mn orbitals, as is shown by the DOS in Figure 2(IIIb).

## 4. Conclusions

The effects of V_O_ and V_Mn_ monovacancies on the structural, electronic, and magnetic properties of the1T–MnO_2_/graphene heterostructure were studied using the GGA approximation within the framework of DFT. We found that the values of formation energy for the heterostructure without and with vacancies of V_O_ and V_Mn_ were −20.81 $\frac{\mathrm{meV}}{{Å}^{2}}\;$, 20.99 $\frac{\mathrm{meV}}{{Å}^{2}}\;$ and −32.11 $\frac{\mathrm{meV}}{{Å}^{2}}$. These values are negative; therefore, the three heterostructures are energetically stable. We found the presence of V_O_ and V_Mn_ monovacancies in the heterostructure induce: (a) a slight decrease in the interlayer separation distance in the 1T–MnO_2_/graphene heterostructure of ~0.13% and ~1.41%, respectively, and (b) a contraction of the bond lengths (Mn-O) of the neighbor atoms of the V_O_ and V_Mn_ monovacancies of ~2.34% and ~6.83%, respectively.

The calculations of the Bader charge for the heterostructure without and with V_O_ and V_Mn_ monovacancies show that these monovacancies induce significant changes in the charge of the first neighbor atoms of the V_O_ and V_Mn_ vacancies, generating chemically active sites (locales) that could improve the adsorption of external atoms (for example, lithium) and molecules. Hence, the heterostructure with V_O_ and V_Mn_ vacancies has potential application for the removal of environmental contaminants, as anode material for lithium-ion batteries, and for energy storage in batteries. Finally, from the analysis of the band structure and the DOS, we found that for the heterostructure without vacancies and V_O_ vacancy, the Dirac cone of the graphene remains intact, while the 1T–MnO_2_ monolayer in the heterostructure with and without V_O_ vacancy exhibits a half-metallic behavior that comes from the hybridization between the 3d–Mn and 2p–O states. In both cases, the heterostructure with and without V_O_ vacancies has a magnetic moment of 3.00 *μ_β_*/Mn, because of which the 1T–MnO_2_/graphene heterostructure without and with V_O_ vacancies could have potential applications in spintronics.

## Figures and Tables

**Figure 1 materials-15-02731-f001:**
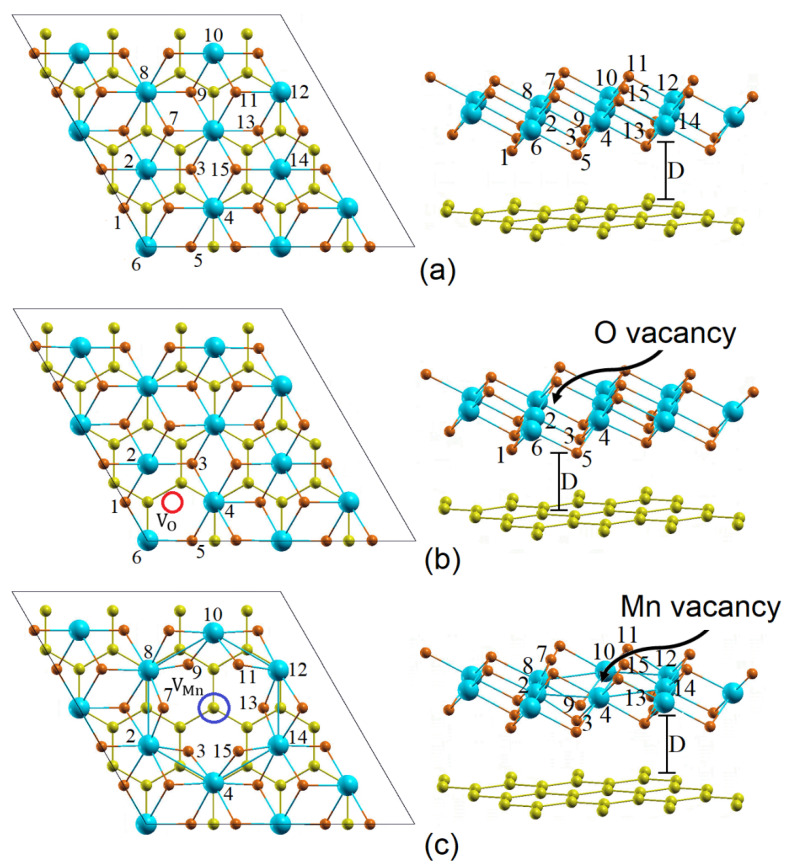
Top and side views of the T–${\mathrm{Mn}}^{12-x}O_{2}^{24-y}$ ($2\sqrt{3}\times 2\sqrt{3}$)/graphene (4×4) interface. (**a**) Pure (x=0 and y=0), (**b**) V_O_ (x=0 and y=1), and (**c**) V_Mn_ (x=1 and y=0). The blue, orange, and yellow spheres represent the Mn, O, and C atoms, respectively.

**Figure 2 materials-15-02731-f002:**
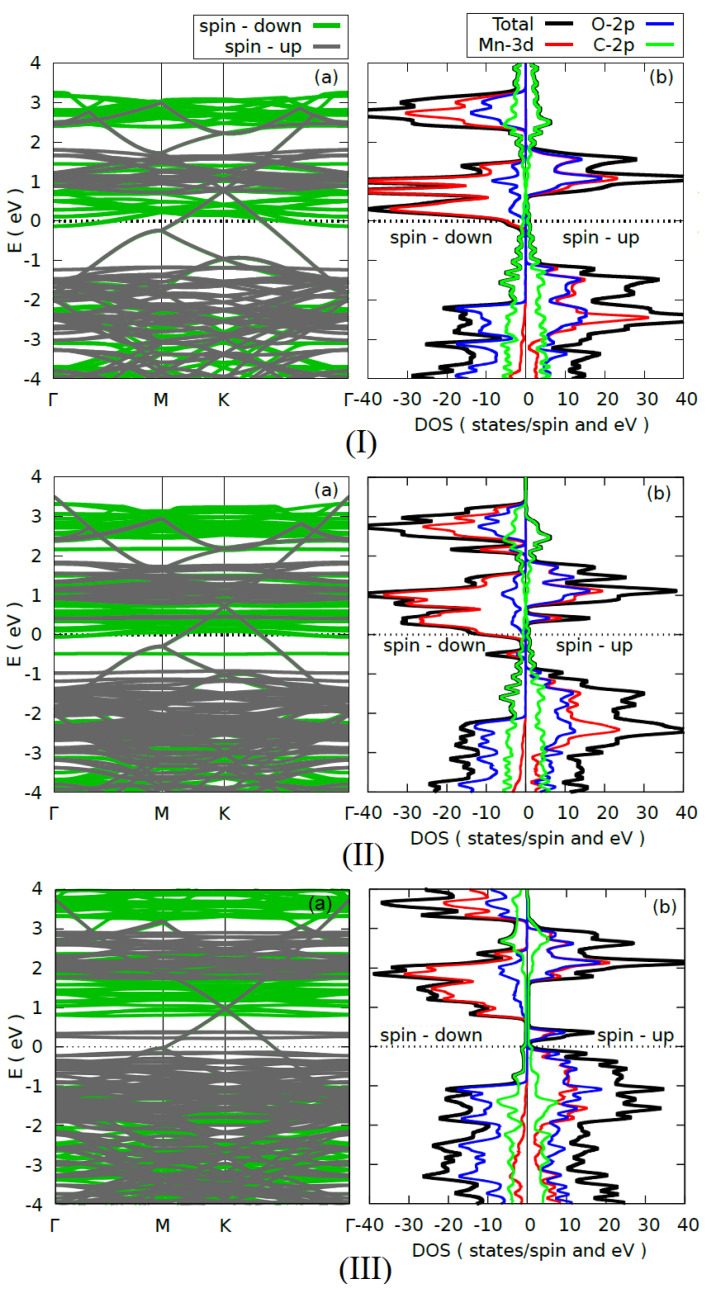
Electronic structure: (**a**) diagram of the energy bands and (**b**) DOS of the (**I**) T–MnO_2_/graphene without a vacancy, (**II**) T–MnO_2_/graphene with V_O_, and (**III**) T–MnO_2_/graphene with V_Mn_. The dotted line represents the Fermi level.

**Table 1 materials-15-02731-t001:** Bond lengths (in Å), for the atoms closest to V_O_ and interfacial distance D (in Å).

Heterostructure	*l* _1-2_	*l* _2-3_	*l* _3-4_	*l* _4-5_	*l* _5-6_	D
Without vacancies	1.917	1.917	1.916	1.916	1.916	2.981
With V_O_	1.875	1.875	1.875	1.875	1.874	2.977

**Table 2 materials-15-02731-t002:** Bond lengths (in Å), for the atoms closest to V_Mn_ and interfacial distance D (in Å).

Heterostructure	*l* _7-8_	*l* _8-9_	*l* _9-10_	*l* _10-11_	*l* _11-12_	*l* _12-13_	*l* _13-14_	*l* _14-15_	*l* _15-4_	*l* _2-7_	*l* _3-4_	D
Without vacancies	1.910	1.916	1.917	1.910	1.910	1.916	1.917	1.910	1.910	1.910	1.916	2.981
With V_Mn_	1.788	1.802	1.799	1.786	1.788	1.802	1.799	1.787	1.788	1.787	1.802	2.939

**Table 3 materials-15-02731-t003:** Changes in the Bader charge at the T–MnO_2_/graphene-V_O_ interface.

Neighbors	Atoms *i*	$\Delta Q_{i}(e)$
First	2	+0.214
	4	+0.213
	6	+0.205
Second	1	+0.053
	3	+0.047
	5	+0.048

**Table 4 materials-15-02731-t004:** Changes in the Bader charge at the T–MnO_2_/graphene-V_Mn_ interface.

Neighbors	Atoms *i*	$\Delta Q_{i}^{\prime }(e)$
First	3	−0.157
	7	−0.188
	9	−0.156
	11	−0.189
	13	−0.150
	15	−0.189
Second	2	−0.064
	4	−0.067
	8	−0.064
	10	−0.067
	12	−0.063
	14	−0.061

## Data Availability

The data obtained in this research are unpublished and are not listed in any databases.

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
