# Peer review of "Effects of Mono-Vacancies of Oxygen and Manganese on the Properties of the MnO2/Graphene Heterostructure"

_materials, 2022, doi:10.3390/ma15082731_

Round 1

Reviewer 1 Report

I have reviewed this manuscript titled: “Effects of mono-vacancies of O and Mn on the properties of the 2 T-MnO2/graphene heterostructure”. The authors have conducted a well designed study. The manuscript fits well within the scope of the journal. Author should address following comments to improve the contents

Title is clear however using to many abbreviations.

The introduction is well documented and to the point.

Line 76: “computational” can be deleted as there is not need.

Lines 77-79: Authors are suggested to add some brief details about the density functional theory 77 (DFT) [46,47] and The Kohn-Sham (KS) equations were solved using the Quantum-ESPRESSO 78 computational code [48–50], so that the reader does not have to go back to the cited references.

Results are described comprehensively; however, there is least discussion of result as this is the main weakness, authors are advised to add a section on discussion and critically reflect on results, methods used and limitations.

 Conclusion section should be condensed to keep information focused.

Reviewer 2 Report

I think that the results are interesting, and are recommended for publication after the following modifications.
1. The writing organization is good but the writing style (more explanation of the results in text) needs to be polished.
2. What is the robustness of the proposed method?
3. Write about the advantages of the suggested method over other existing methods?
4. The originality of the paper needs to be stated clearly. It is of importance to have sufficient results to justify the novelty of a high-quality journal paper.
5. The Introduction should make a compelling case for why the study is useful along with a clear statement of its novelty or originality by providing relevant information and providing answers to basic questions such as: What is already known in the open literature? What is missing (i.e., research gaps)? What needs to be done, why and how? Clear statements of the novelty of the work should also appear briefly in the Abstract and Conclusions sections.

Reviewer 3 Report

Dear Authors

This manuscript is focused on the effects of monovacancies of oxygen (VO) and manganese (VMn) on the structural and electronic properties of the 1T-MnO2/graphene heterostructure. The manuscript presented concerns an interesting and actual subject. The following suggestion and comments should be taken:

  1. The overall English needs to be improved. Please seek guidance from a native English speaker if possible ("the" "a", commas, plural form and others could be corrected).
  2. The introduction section needs enhancement 1-3 sentences about calculations of other carbons for enhancement. Please cite (1) Physica B: Condensed Matter 541, (2018), 6-13, https://doi.org/10.1016/j.physb.2018.04.023 (2) Bull Mater Sci 41, 76 (2018). https://doi.org/10.1007/s12034-018-1603-5 (3) Surface Science 711, 121876, (2021) https://doi.org/10.1016/j.susc.2021.121876
  3. Figure 1. Please correct this image for better quality.
  4. Could the authors include the standard deviation of the used methods?
  5. Please add information about basis set and their influence on final results.
  6. Authors are suggested to enhance the conclusions (313 - applications please add more 1-2 sentences or future plans).
  7. Why the authors do not use carbon ends saturated with hydrogen atoms in carbon materials? Please explain in the comments.
  8. It is suggested to perform calculations for larger graphene systems to investigate the dependence on structure size.

Round 2

Reviewer 1 Report

Many thanks for revising the manuscript. 

Reviewer 3 Report

The authors have addressed all comments and the manuscript can be published as is.